# The Global Impact of COVID-19 on Threat Appraisals

**DOI:** 10.3390/healthcare10091718

**Published:** 2022-09-08

**Authors:** Camilla Mattiuzzi, Giuseppe Lippi

**Affiliations:** 1Service of Clinical Governance, Provincial Agency for Social and Sanitary Services (APSS), 38123 Trento, Italy; 2Section of Clinical Biochemistry and School of Medicine, University of Verona, 37126 Verona, Italy

**Keywords:** SARS-CoV-2, COVID-19, anxiety, distress, fear, rumination, worry

## Abstract

We planned an infodemiological analysis to estimate the global impact of coronavirus disease 2019 (COVID-19) on threat appraisals. We accessed Google Trends using the search terms “Anxiety”, “Distress”, “Fear”, “Rumination”, “Stress” and “Worry” within the “topic” domain, setting the geographical location to “worldwide”, between July 2017 and July 2022. The weekly Google Trends score for the six search terms, thus, mirroring Web popularity and probable prevalence, was compared between the two search periods, “pre-COVID” (between July 2017 and February 2020) and COVID (between March 2020 and July 2022), thus, reflecting the volume of searches before and during the ongoing COVID-19 pandemic. The median weekly Google Trends score of all these search terms significantly increased during the COVID-19 pandemic, i.e., anxiety by 22%, distress by 13%, fear by 9%, rumination by 18%, stress by 13% and worry by 20%. With variable strength, the weekly Google Trends scores of each search term were found to be significantly associated (all *p* < 0.001). We can, hence, conclude that the enhanced burden of threat appraisals observed after SARS-CoV-2 spread leads the way to establish preventive, diagnostic and therapeutic measures in order to limit the unfavorable mental health consequences caused by the ongoing COVID-19 pandemic.

## 1. Introduction

Coronavirus disease 2019 (COVID-19), first identified at the end of 2019 in the Chinese city of Wuhan [1], has now become a pandemic infectious disease, affecting over 570 million people and causing nearly 6.4 million deaths at the end of July 2022, according to the World Health Organization (WHO) Coronavirus (COVID-19) Dashboard [2]. The pathology, sustained by a novel beta coronavirus called severe acute respiratory syndrome coronavirus 2 (SARS-CoV-2) by the Coronaviridae Study Group of the International Committee on Taxonomy of Viruses [3], may cause a kaleidoscope of injuries and functional impairments in the lungs as well as in many other organs and tissues, which may persist in a significant number of affected subjects, even after recovering from the acute infection [4].

In addition to the risk of developing severe and occasionally persistent organic derangements, the ongoing COVID-19 pandemic is also apparently enhancing the threat appraisals of both affected and unaffected individuals. Several recent studies, involving populations of all ages, from different geographical locations and not necessarily engaged in patient care, have, in fact, reported an enhanced burden of psychological disturbances, including anxiety and stress, namely due to concern/fear of being infected and developing acute and/or long-term unfavorable consequences of COVID-19 [5,6,7,8]. Infodemiology, frequently used as a synonym of digital epidemiology, is currently regarded as a mainstay of general medical knowledge acquisition [9], as well as a potentially powerful tool to face the ongoing COVID-19 pandemic [10]. In particular, recent evidence has been provided that innovative data streams, such as those garnered from Google Trends, may support healthcare experts and policymakers in planning strategic priorities and defining proactive policies to mitigate and tackle a kaleidoscope of COVID-19-related disturbances, including mental health problems [11,12]. In order to more clearly acknowledge how the worldwide population is behaving and coping during a highly stressful situation, such as the ongoing SARS-CoV-2 pandemic, we, hence, planned an infodemiological analysis to estimate the global impact of COVID-19 on threat appraisals, defined as a combination of perceived severity (i.e., harm degree) and vulnerability (the risk of experiencing harm) related to a specific ongoing circumstance [13].

## 2. Materials and Methods

We accessed Google Trends (Google Inc., Mountain View, CA, USA) using the search terms “Anxiety”, “Distress”, “Fear”, “Rumination”, “Stress” and “Worry” on 1 August 2022, within the “topic” domain (and thereby overcoming potential language barriers), setting the geographical location to “worldwide” and limiting our analysis to the past 5 years (i.e., from July 2017 to July 2022). These search terms were selected as being largely used in previous publications to define unfavorable post-COVID-19 mental health outcomes [5,6,7,8]. The weekly Google Trends score for the search terms, thus, reflecting their Web popularity and probable prevalence, was downloaded into a Microsoft Excel file (Microsoft, Redmond, WA, USA). We classified the search period as “pre-COVID” (between July 2017 and February 2020) and COVID (between March 2020 and July 2022). We identified the cut-point between the two periods in March 2020, as this corresponds to the official WHO declaration of the pandemic, after which threat appraisals in the population would have become more prevalent. The weekly Google Trends scores of each keyword, thus, mirroring their volume of Google searches, were reported as median and interquartile range (IQR) and were analyzed using Spearman’s correlation (with 95% confidence interval; 95%CI) and Mann–Whitney test, using Analyse-it (Analyse-it Software Ltd., Leeds, UK).

The study was conducted in accordance with the Declaration of Helsinki, under the terms of relevant local legislation. This analysis was based on electronic searches in an open and publicly available repository (Google Trends) and, thus, no informed consent or Ethical Committee approval were necessary.

## 3. Results

The results of this infodemiological analysis are shown in Figure 1.

The median weekly Google Trends score of all the investigated domains significantly increased during the COVID-19 pandemic. Specifically, anxiety increased by 22% from 64 (IQR, 61–67) to 78 (IQR, 75–81; *p* < 0.001), distress by 13% from 60 (IQR, 55–63) to 68 (IQR, 62–75; *p* < 0.001), fear by 9% from 73 (IQR, 71–76) to 80 (IQR, 76–83; *p* < 0.001), rumination by 18% from 57 (IQR, 53–61) to 67 (IQR, 64–71), stress by 13% from 76 (IQR, 70–79) to 86 (IQR, 82–89; *p* < 0.001), and worry by 20% from 64 (IQR, 58–67) to 77 (IQR, 74–85; *p* < 0.001). The Spearman’s correlation between the median weekly Google Trends score of all these search terms over time is summarized in Table 1.

With variable strength, the weekly Google Trends scores of each search term were found to be significantly associated (all *p* < 0.001), with the highest correlation found between the Google search volumes of stress and worry (r = 0.81), followed by anxiety and rumination (r = 0.80), anxiety and worry (r = 0.80), distress and worry (r = 0.80). The weakest correlation was found between the Google Trends scores of fear and rumination (r = 0.57), as well as those of fear and stress (r = 0.62).

## 4. Discussion

Important evidence has emerged from several clinical trials, recently meta-analyzed by Lekagul and colleagues [14], revealing that the accurate estimation of the burden of COVID-19-related threat appraisals is essential for establishing effective measures for mitigating the psychological consequences of the ongoing pandemic on COVID-19 patients, healthcare workers as well as within the general population. We, hence, planned this infodemiological analysis for attempting to estimate the global impact of COVID-19 on threat appraisals.

Although no sub-analysis could be carried out since the searches in Google Trends cannot be more specifically driven to cluster demographical information, our findings reveal that COVID-19 may have imposed a considerable mental health impact all around the world, as attested by the significantly increased volume of worldwide Google searches for terms, such as “Anxiety”, “Distress”, “Fear”, “Rumination”, “Stress” and “Worry”, after March 2020, when the WHO declared COVID-19 a pandemic (Figure 1) [15]. More specifically, the weekly Google Trends score of anxiety, rumination and worry increased by around 20% compared to the pre-COVID-19 period, whilst the volume of Google searches for distress, stress and fear increased by around 10%. Interestingly, the statistically significant association found between the weekly Google Trends score of each of these search terms over time suggests that many and potentially unrelated psychological domains (i.e., fear, worry, rumination) may have been concomitantly affected by COVID-19, whilst the persistence over time of such derangements remains rather unpredictable and unquestionably concerning as yet. This evidence shall lead the way to a reinforced recognition of unfavorable post-COVID-19 mental health outcomes, which may foster an increased burden of psychological consultations in forthcoming years. Accordingly, there will probably be a compelling need to readapt existence structures or even develop new outpatient facilities for psychological counselling of fragile people who may bear the mental consequences of the pandemic for many years. Additional strategies could also be urgently planned to face an increased burden of threat appraisals, such as changes in consultation practice, by implementing or potentiating online counselling [16].

## 5. Conclusions

The evidence that the threat appraisals may have dramatically increased after the spread of SARS-CoV-2 calls for an urgent need to establish preventive, diagnostic and therapeutic measures, such as those previously mentioned, in order to limit the unfavorable mental health consequences caused by the ongoing COVID-19 pandemic.

## Figures and Tables

**Figure 1 healthcare-10-01718-f001:**
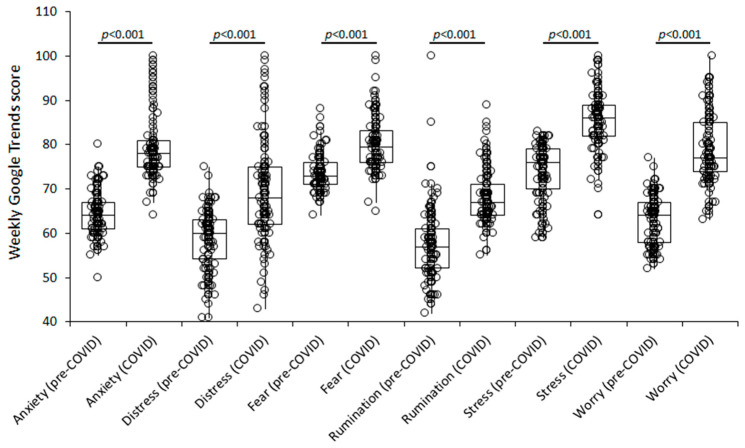
Worldwide weekly Google Trends score of the search terms “Anxiety”, “Distress”, “Fear”, “Rumination”, “Stress” and “Worry” before (“pre-COVID”) and during (“COVID”) the coronavirus disease 2019 (COVID-19) pandemic. Results are shown as median and interquartile range (IQR).

**Table 1 healthcare-10-01718-t001:** Spearman’s correlation (and 95% confidence interval; 95%CI) between the median weekly Google Trends score of the search terms “Anxiety”, “Distress”, “Fear”, “Rumination”, “Stress” and “Worry” during the past 5 years (i.e., July 2017–July 2022).

Search Term	Distress	Fear	Rumination	Stress	Worry
Anxiety	0.66 (0.59 to 0.72; *p* < 0.001)	0.65 (0.58 to 0.72; *p* < 0.001)	0.80 (0.76 to 0.84; *p* < 0.001)	0.78 (0.73 to 0.82; *p* < 0.001)	0.80 (0.75 to 0.84; *p* < 0.001)
Distress	-	0.67 (0.60 to 0.73; *p* < 0.001)	0.67 (0.60 to 0.73; *p* < 0.001)	0.75 (0.69 to 0.79; *p* < 0.001)	0.80 (0.75 to 0.84; *p* < 0.001)
Fear	-	-	0.57 (0.48 to 0.65; *p* < 0.001)	0.62 (0.54 to 0.69; *p* < 0.001)	0.77 (0.71 to 0.81; *p* < 0.001)
Rumination	-	-	-	0.69 (0.62 to 0.75; *p* < 0.001)	0.71 (0.65 to 0.77; *p* < 0.001)
Stress	-	-	-	-	0.81 (0.77 to 0.85; *p* < 0.001)

## Data Availability

Data will be available upon reasonable request to the corresponding author.

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
