# Peer review of "The Global Impact of COVID-19 on Threat Appraisals"

_healthcare, 2022, doi:10.3390/healthcare10091718_

Round 1

Reviewer 1 Report

I would like to thank the authors for presenting an interesting report that can attract the readers' attention. As a brief report, I have a few comments for the authors to consider:

1. The authors could move the introduction of infodemiology (currently in the beginning of the Discussion section) to the Introduction section. This is for readers who don't know this term so that a brief understanding can be acquired first. 

2. The authors need to briefly explain in Section 2 why the search terms (e.g., anxiety and distress) were selected, and what each term means (I believe there are differences between them).

3. The authors need to explain why they used Feb. 2020 as the cut-off as they said in the Introduction COVID-19 started at the end of 2019. (They also mentioned in the Discussion that this was because of the WHO's declaration. If this was the case, it should be mentioned earlier).

4. The authors are suggested to provide brief implications for their results in the Discussion. For example, the authors mentioned preventative, diagnostic and therapeutic measures. Can they be more concrete so that policymakers can be better informed by this report? Otherwise, it could be naturally perceived by readers that people experienced more mental health problems during the pandemic. The Conclusion section can benefit from this as well.

5. The authors are suggested to provide a data statement, or attach the file as an appendix so that it can be replicated.

6. Please define 'threat appraisals'.

7. In line 47, if I understand it correctly, should it be 'six search terms' rather than 'three search terms'? Or the authors can clearly say what the three terms are.

Hope these are helpful.

Author Response

attention. As a brief report, I have a few comments for the authors to consider:

  • We are thankful to the referee for the globally favourable comments on our manuscript. We’ll do our best to improve it according to the referee’s suggestions.

  1. The authors could move the introduction of infodemiology (currently in the beginning of the Discussion section) to the Introduction section. This is for readers who don't know this term so that a brief understanding can be acquired first.
  • ANSWER: Very good point, thanks. Done, as suggested.

  1. The authors need to briefly explain in Section 2 why the search terms (e.g., anxiety and distress) were selected, and what each term means (I believe there are differences between them).
  • ANSWER: Good point, thanks. Paper modified as follows: “These search terms have been selected as being largely used in previous publications to define to define unfavorable post-COVID mental health outcomes [5-8]” (the pertinent references have been cited). Then, technically speaking, we select both “anxiety” and “distress” since they would express different meaning as in the Cambridge dictionary: distress: “a feeling of extreme worry, sadness, or pain” (https://dictionary.cambridge.org/dictionary/english/distress); anxiety: “an uncomfortable feeling of nervousness” (https://dictionary.cambridge.org/dictionary/english/anxiety). Hope this explanation will actually reply to the important referee’s question.

  1. The authors need to explain why they used Feb. 2020 as the cut-off as they said in the Introduction COVID-19 started at the end of 2019. (They also mentioned in the Discussion that this was because of the WHO's declaration. If this was the case, it should be mentioned earlier).
  • ANSWER: Good point thanks. We have indeed clarify this matter (exactly as supposed by the referee), as follows: “We identified the cut-point between the two periods in March 2022, as this corresponds to the official WHO declaration of the pandemic, after which threat appraisals in the population would have become more prevalent”.

  1. The authors are suggested to provide brief implications for their results in the Discussion. For example, the authors mentioned preventative, diagnostic and therapeutic measures. Can they be more concrete so that policymakers can be better informed by this report? Otherwise, it could be naturally perceived by readers that people experienced more mental health problems during the pandemic. The Conclusion section can benefit from this as well.
  • ANSWER: This is a very good point. Paper revised accordingly, as follows: “This evidence shall lead the way to a reinforced recognition of the unfavorable post-COVID mental health outcomes, which may foster an increased burden of psychological consultations in the forthcoming years. Accordingly, there will probably be a compelling need of readapting existence structures, or even developing new outpatient facilities, for psychological counselling of fragile people who may bear for long the mental consequences of the pandemic. Additional strategies could also be urgently planned to face an increased burden of threat appraisals, such as changes in consultation practice, by implementing or potentiating online counselling [16]”. A new reference [#16 has been included].

  1. The authors are suggested to provide a data statement, or attach the file as an appendix so that it can be replicated.
  • ANSWER: Done, as follows: “Data Availability Statement: Data will be available upon reasonable request to the corresponding author”.

  1. Please define 'threat appraisals'.
  • ANSWER: Very good point thanks. Done, as follows: “defined as combination of perceived severity (i.e., harm degree) and vulnerability (the risk of experiencing harm) related to a specific ongoing circumstance [13]”. The pertinent (new #13) citation has been included.

  1. In line 47, if I understand it correctly, should it be 'six search terms' rather than 'three search terms'? Or the authors can clearly say what the three terms are..
  • ANSWER: Yes, this is absolutely correct. Apologizes for this typing mistake. Text corrected accordingly.

Reviewer 2 Report

Greetings and Regards
While thanking the authors of the article for reviewing this issue, I suggest the following things to improve and strengthen this article:
The abstract is not attractive and should be complete and explain the method of data collection.
Introduction: It is very weak and the importance of the topic and the reason for addressing this issue is not clear. Use new findings. State the need to address this issue. Write down the problem that you intend to solve with this research.
The methodology needs to be revised: the research method should be written. Statistical population, statistical sample and sampling method should be stated. Data collection tools and measurement scales should be introduced. Add how to check the validity and reliability of the tool. How long have you been collecting data? Was it at the peak of the virus outbreak or not? Explain the method of data analysis.
Findings: You should add demographic findings. Is it clear whether the samples or their relatives have been infected with the virus during this period or not? Or how much they have been exposed to disappointing media news? These affect the results.
Discussion and conclusion: It is weak and should be strengthened. Add theoretical foundations, existing problems, research findings as well as background research to strengthen the discussion and conclusions.
Finally, based on the research findings, give practical suggestions from a psychological point of view and recommend responsible organizations.
Use new resources.
Good luck

Author Response

While thanking the authors of the article for reviewing this issue, I suggest the following things to improve and strengthen this article:

  • We are thankful to the referee for the globally favourable comments on our manuscript. We’ll do our best to improve it according to the referee’s suggestions.

The abstract is not attractive and should be complete and explain the method of data collection.

  • ANSWER: Good point, thanks. We have re-edited the abstract as suggested, as follows: “We accessed Google Trends using the search terms “Anxiety”, “Distress”, “Fear”, “Rumination”, “Stress” and “Worry” within the “topic” domain, setting the geographical location to “worldwide”, between July 2017 and July 2022. The weekly Google Trends score for the six search terms, thus mirroring Web popularity and probable prevalence, was compared between the two search periods “pre-COVID” (between July 2017 and February 2020) and COVID (between March 2020 and July 2022), thus reflecting the volume of searches before and during the ongoing COVID-19 pandemic”.

Introduction: It is very weak and the importance of the topic and the reason for addressing this issue is not clear. Use new findings. State the need to address this issue. Write down the problem that you intend to solve with this research.

  • ANSWER: Good point, thanks. The introduction has been reformulated, taking also into account the suggestion of the first referee (new changes are all highlighted in yellow throughout the introduction). We could not prolong too much this part for reasons of word-count (this paper is submitted a “brief note”).

The methodology needs to be revised: the research method should be written. Statistical population, statistical sample and sampling method should be stated. Data collection tools and measurement scales should be introduced. Add how to check the validity and reliability of the tool. How long have you been collecting data? Was it at the peak of the virus outbreak or not? Explain the method of data analysis.

  • ANSWER: Replies as follows:
    • The research method has been well-defined in the section “methods”, as follows: “We accessed Google Trends (Google Inc. Mountain View, CA, US) using the search terms “Anxiety”, “Distress”, “Fear”, “Rumination”, “Stress” and “Worry” on August 1, 2022, within the “topic” domain (and thereby overcoming potential language barriers), setting the geographical location to “worldwide”, and limiting our analysis to the past 5 years (i.e., from July 2017 to July 2022).”
    • The period of data collection has been stated (“August 1, 2022”).
    • As concerns “Statistical population, statistical sample and sampling method should be stated”, this part cannot be applied to our analysis. There is no patient sample (the search is “worldwide” and thereby includes all the world population, so that neither a sampling method has been (or could be) used.
    • As concerns “Was it at the peak of the virus outbreak or not?”, collection has been carried out over a 5-year period (see definition of the two search periods in the text), and thereby comprehends all the COVID period and a similar period o0f nearly 2.5 years before (this is all clearly stated in the “methods” section).

Findings: You should add demographic findings. Is it clear whether the samples or their relatives have been infected with the virus during this period or not? Or how much they have been exposed to disappointing media news? These affect the results.

  • ANSWER: We are sorry, but in this “worldwide” (brief) analysis no demographical data could be obtained, nor the search in Google Trends could be more specifically driven to cluster information according to demographical variables. This aspect has been included as a limitation in our work.

Discussion and conclusion: It is weak and should be strengthened. Add theoretical foundations, existing problems, research findings as well as background research to strengthen the discussion and conclusions.

  • ANSWER: Very good point, thanks. The entire discussion has been reformulated, taking into account the suggestion of the referee, and thus also providing some suggestion on how to copy with this evidence.

Finally, based on the research findings, give practical suggestions from a psychological point of view and recommend responsible organizations. Use new resources.

  • ANSWER: Done, as follows: “The evidence that the threat appraisals may have dramatically increased after the spread of SARS-CoV-2 calls for an urgent need to establish preventive, diagnostic and therapeutic measures in order limit the unfavorable mental health consequences caused by the ongoing COVID-19 pandemic”.